# Evaluating Morphological Alignment of Tokenizers in 70 Languages

**Catherine Arnett** [1]  **Marisa Hudspeth** [2]  **Brendan O'Connor** [2]

## Abstract

While tokenization is a key step in language modeling, with effects on model training and performance, it remains unclear how to effectively evaluate tokenizer quality. One proposed dimension of tokenizer quality is the extent to which tokenizers preserve linguistically meaningful subwords, aligning token boundaries with morphological boundaries within a word. We expand MorphScore (Arnett & Bergen, 2025), which previously covered 22 languages, to support a total of 70 languages. The updated MorphScore offers more flexibility in evaluation and addresses some of the limitations of the original version. We then correlate our alignment scores with downstream task performance for five pre-trained languages models on seven tasks, with at least one task in each of the languages in our sample. We find that morphological alignment does not explain very much variance in model performance, suggesting that morphological alignment alone does not measure dimensions of tokenization quality relevant to model performance.

 MorphScore evaluator
 datasets  code and data

## 1. Introduction

Tokenization is the first step of language modeling, in which strings of text are segmented into discrete units in the tokenizer's vocabulary. Tokenization has been shown to have effects on speed and efficiency of language model training (Dagan et al., 2024; Ali et al., 2024; Asgari et al., 2025), performance (Ali et al., 2024), and inference cost and latency (Ahia et al., 2023; Petrov et al., 2023). Despite this, it is still unclear how to best evaluate tokenizers. Finding reliable intrinsic tokenizer evaluation would be enormously valuable,

as it would enable tokenizer selection before model training, leading to significant computational and financial savings.

One of the most frequently used intrinsic tokenizer evaluations is compression. Compression is often measured as the number of tokens it takes to encode a text given a particular tokenizer. It is relatively easy to measure, as it requires simply tokenizing a text and calculating token counts. One metric of compression is fertility, i.e. the number of tokens per word (Rust et al., 2021). Fertility is simple to implement but can be difficult to generalize crosslinguistically, as wordhood is often operationalized as whitespace-separated orthographic units. Not all languages use whitespaces, e.g. Mandarin Chinese, Thai, and Khmer. Corpus token count (CTC; Schmidt et al., 2024) is the total tokens it takes to represent a text for a given tokenizer. CTC can be compared, therefore across tokenizers of different types, vocabulary sizes, etc. It has also been used to compare compression crosslinguistically, by calculating CTC over parallel text in order to determine crosslinguistic differences in compression (Arnett & Bergen, 2025).

Some have argued that increased compression increases the information density for a sequence of fixed length, which could lead to improved model performance (Deletang et al., 2024). There has been empirical evidence to support the claim that more tokenizer compression is correlated with better task performance (Goldman et al., 2024; Gallé, 2019). However, more recent work has shown that there is no robust relationship between tokenizer compression and language model performance (Schmidt et al., 2024).

Other intrinsic tokenizer evaluations have been proposed, such as Rényi efficiency (Zouhar et al., 2023), which takes into account frequency distribution. More optimal Rényi efficiency is associated with having more compression for higher-frequency items and less compression for lower-frequency items. Zouhar et al. (2023) released the `tokenization-scorer` package to support calculation of Rényi efficiency for any tokenized text. However, later work argues it may not provide a holistic metric of good tokenization quality (Cognetta et al., 2024).

Another property of tokenizers that has been studied is how morphologically aligned tokenization is, or to what extent do token boundaries align with morpheme boundaries for a given word. For example, the English word 'books' is

---

[1]EleutherAI [2]University of Massachusetts Amherst. Correspondence to: Catherine Arnett <catherine@eleuther.ai>.

*Proceedings of the ICML 2025 Tokenization Workshop (TokShop)*, Vancouver, Canada. PMLR 267, 2025. Copyright 2025 by the author(s).

composed of the stem 'book' and the plural suffix '-s'. The morphologically aligned segmentation would be [book + s]. Non-aligned segmentations include [boo + ks] or [bo + oks].

There are several studies which argue that morphologically aligned tokenization is associated with improved performance on a variety of NLP tasks (Park et al., 2020; Vasiu & Potolea, 2020; Bostrom & Durrett, 2020; Hofmann et al., 2021; Nzeyimana & Niyongabo Rubungo, 2022; Erkaya, 2022; Toraman et al., 2023; Drík & Forgac, 2024; Libovický & Helcl, 2024; Jabbar, 2024; Uzan et al., 2024; Bauwens & Delobelle, 2024; Asgari et al., 2025). Despite the volume of work on this topic, it is still difficult to conclude whether morphological alignment of tokenizers *generally* improves downstream performance. Prior work varies widely in language coverage, model architectures, amount of supervision (zero shot through full supervised finetuning), and evaluation metrics (e.g. perplexity versus performance on various downstream tasks).

Batsuren et al. (2024) developed an evaluation in which the tokenization of a given word was classified according to whether words were split into morphemic tokens or non-morphemic tokens, or were stored whole as a single token. The authors found that morphemic tokenization was correlated with better performance. MorphScore (Arnett & Bergen, 2025) expands on this idea and measures how often tokenizer boundaries align with morpheme boundaries for 22 languages. However, the authors found that MorphScore was not predictive of model performance (Arnett & Bergen, 2025). Arnett et al. (2024) found that morphemic tokenization had only a small effect on performance at a subject-verb agreement task in Spanish. There is also evidence from a variety of different languages that morphologically aligned tokenization did not benefit model performance (Macháček et al., 2018; Saleva & Lignos, 2021; Choo & Kim, 2023).

The original MorphScore is limited, however. While relatively diverse, the language coverage does not include many high-resource languages that are commonly represented in language model research, e.g. French or German. There are also design choices in the creation of MorphScore that limit its potential utility. The items in MorphScore do not have any context. While this does not impact tokenization which uses whitespace pre-tokenization, this makes it impossible to accurately evaluate morphological alignment of superword tokenizers, e.g. SuperBPE (Liu et al., 2025) and BoundlessBPE (Schmidt et al., 2025). Other information from the Universal Dependencies (UD), which were used to create MorphScore was also not included, such as part-of-speech (POS) information or morphological information.

MorphScore also does not take into consideration item frequency. As discussed in Zouhar et al. (2023), optimal tokenization may be dependent on frequency distribution. It may be more important for tokenization of more frequent items to be morphologically aligned, as they occur more often. Or, it may be more important for low-frequency items to be tokenized morphemically, as lower-frequency words are more likely to be segmented into multiple tokens using popular tokenization algorithms like Byte-Pair Encoding (BPE; Gage, 1994; Sennrich et al., 2016).

Our updated and expanded MorphScore, allows us to better determine under which settings morphologically aligned tokenization contributes to better model performance. Given the mixed evidence in previous work, additional analyses with broader language coverage is necessary. We expand MorphScore to cover 70 languages. This version of MorphScore allows the user to set parameters, such as including frequency information and the scoring of single-token words.

In this paper, we test the effects of these different parameter settings on the morphological alignment scores. Our datasets also include sentential context, POS information, and the morphological information included in UD. While we do not analyze these factors here, we include them in order to enable a broad range of future work.

## 2. Creating Evaluation Datasets

**Data.** All datasets are built using the annotations from Universal Dependencies[1]. The exact treebanks we used are listed in Appendix A. For each language, we generally chose the largest available treebank and used all available splits (train, dev, and test). We include the test split, as for many of the languages that is the only split available. We exclude words which are composed of a single morpheme, as there is no morpheme boundary to evaluate on. For each annotated word, we use the wordform and the lemma to determine a proposed segmentation. For example, for the wordform 'launched', the provided lemma is 'launch'. Therefore, by identifying the longest shared sequence between the wordform and lemma, we determine 'launch' to be the stem and '-ed' to be the affix. Any preceding and subsequent characters are treated as the prefix and suffix, respectively. Thus, the gold segmentation will have at least two morphemes (the stem and an affix) and at most three morphemes (a prefix, stem, and suffix).

As in the original MorphScore, we only select cases where there the wordform can be recomposed by concatenating the proposed stem and the affixes, in order to remove irregular forms and examples of non-concatenative morphology, where determining a gold segmentation is less straightforward. Therefore, we used only examples where the identified stem did not undergo suppletion, umlaut, etc., and the wordform could be composed of the stem and either a prefix, as suffix, or both. We observe that without this crite-

---

[1] https://universaldependencies.org/

Arabic root k-t-b (ك-ت-ب)

(a) كَتَبَ
kataba
'he wrote'

(b) كاتِب
kātib
'writer'

*Figure 1.* Example of root template pattern in Arabic.

rion, we could get gold segmentations that would not be informative about the quality of tokenization. For example, the infinitival form of the verb 'to be' in Afrikaans is *wees*. The present form for all persons and numbers is *is*. Under our segmentation approach, the stem would be identified as -s and the proposed gold segmentation would be [i + s]. However, *is* is an irregular form and it should not be thought of as having the stem -s.

In the process of creating and filtering the datasets, despite having very large treebanks, there were not sufficient remaining items from any of the Semitic languages (Amharic, Arabic, and Hebrew) or most isolating languages (e.g. Chinese, Vietnamese, and Thai), which are introflexive languages. In these languages, many morphological processes are encoded using non-concatenative morphology. In particular, these languages often use root template patterns, where a group of consonants is used for a series of related words. Changing the intervening vowels changes the meaning, e.g. from verb to noun (cf. *kataba* 'he wrote' and *kātib* 'writer'; Figure 1). Recent work has sought solutions for effective tokenization in languages with these morphological patterns (Gazit et al., 2025).

Isolating languages like Vietnamese and Chinese are not included, because there are not sufficient affixation patterns to create the kind of examples that are selected for by our dataset creation process. In these languages, most words do not have overt morphological markings for number, tense, etc. Therefore, this approach only covers fusional and agglutinative languages. Future work could focus on how to determine gold segmentations for both irregular items, such as the example from Afrikaans, and non-concatenative morphology.

In total, we created datasets for 86 languages. Once our datasets were created, we filtered out languages for which there were fewer than 100 items. This leaves a set of 70 languages. All languages are listed in Appendix A. We release the unfiltered datasets, including those that ultimately had too few examples to be scored, on Hugging Face.[2]

**Scoring.** We expand on MorphScore by incorporating both boundary-level and subword-level evaluations. Specifically, our evaluator calculates:

- macro average *boundary* precision and recall
- micro and macro average *subword* precision, recall, and F1

Boundary metrics evaluate whether the predicted tokenization correctly identifies morpheme boundaries, focusing solely on boundary placement. In contrast, subword metrics assess whether the predicted subword spans exactly match gold morphemes.

For example, if the gold segmentation is [book + s] and the predicted tokens are [boo + k + s], only the boundary between 'k' and 's' is correct. This yields a boundary precision of 1/2 and a boundary recall of 1/1. However, for subword metrics, only the token 's' matches a gold morpheme exactly, resulting in a subword precision of 1/3 and recall of 1/2. The code for running scoring is released on GitHub[3]. We report the individual scores for each language and each tokenizer in Appendix B and full results are released on OSF[4].

**Oversegmentation and Accuracy.** If morphological alignment is measured using accuracy, then a tokenizer can achieve a perfect alignment score by segmenting a word into characters. For example segmenting 'books' into [b + o + o + k + s] leads to an accurate segmentation. This should not be considered a morphologically aligned tokenization. The Llama tokenizers, for several of the languages with non-Latin scripts, tokenize words into tokens more granular than characters, e.g. separating characters and diacritics or decomposing into bytes. Therefore, oversegmentation leads to high accuracy. We find that tokenizing words into more tokens is strongly correlated with morphological alignment as measured with accuracy. In contrast to the original implementation of MorphScore, we use precision and recall as evaluation metrics. Precision, in particular, penalizes tokenizers for oversegmentation.

## 3. Effect of Parameter Settings

Here, we explore the effects of two parameters of the scoring function on alignment score and how they interact with each other. Our goal is to determine the optimal default settings for evaluating morphological alignment.

---

[2] https://huggingface.co/datasets/catherinearnett/morphscore
[3] https://github.com/catherinearnett/morphscore
[4] https://osf.io/eqy64/

**Frequency Scaling.** One parameter we set is whether we weight the morphological alignment score by the wordform frequency, as measured in the UD treebank we used to create the dataset for a given language. Higher-frequency items would be weighted more heavily in the final score than lower-frequency items. Taking frequency distribution into account could lead to a more informative measurement of tokenization quality.

We also test whether there is a correlation between an item's frequency and the likelihood that a tokenizer segments it in a morphologically aligned way. We compute Spearman's rank correlation coefficient across all items and find a weak but statistically significant correlation ($\rho = 0.119$, p $< 0.0001$). The relationship is positive, so more frequent items are more likely to be morphemically segmented.

**One-Token Words.** Next, we test whether there is a difference in scores depending on whether items that are tokenized into a single token are included in the score calculation. If they are included, the tokenization receives the score associated with a morphologically aligned tokenization. One argument for excluding these items is that these cases do not give any indication of how morphologically aligned a segmentation of a word is, given that there is a segmentation. The alignment score can be inflated for languages where it is possible for the tokenizer to store many whole words in its vocabulary. However, excluding these cases might also essentially penalize a tokenizer for segmenting less. Fewer segmentations leads to better compression, which is thought to be an ideal feature of a tokenizer, as discussed above.

We find there is a significant difference based on the inclusion of one-token items. Morphological alignment scores are generally higher with the inclusion of one-token items, which is what we predicted. We also find an interaction between word frequency and the likelihood that a tokenizer represents a word as a single token. This is a feature of most tokenization algorithms. More frequent items are more likely to be stored in the vocabulary, instead of having to be composed of multiple tokens. In an item-wise test, there is a negative correlation between word frequency and the number of tokens a word is segmented into (Spearman's $\rho$ = -0.108, p $< 0.0001$).

**Optimal Default Settings.** We test whether there are differences in morphological alignment scores as we vary frequency scaling and the inclusion of one-token words. We fit a linear mixed effects model with morphological alignment precision as the dependent variable. Frequency scaling, one-token words, and training split are each fixed effects. We test for effects of each of these and their interactions. We include the tokenizer as a random intercept. We report the full statistical results in Appendix C.

Table 1. Morphological alignment of pre-trained tokenizers.

| Tokenizer | Morph. Alignment | |
| --- | --- | --- |
| | **Recall** | **Precision** |
| **BLOOM** | 0.33 ± 0.00 | 0.11 ± 0.00 |
| **Gemma3** | 0.35 ± 0.00 | 0.12 ± 0.00 |
| **Llama2** | **0.56 ± 0.00** | 0.13 ± 0.00 |
| **Llama3** | 0.45 ± 0.00 | 0.12 ± 0.00 |
| **XGLM** | 0.52 ± 0.00 | **0.23 ± 0.00** |

There are significant differences across the different categories. We compare the relative ranks according to precision score for the different conditions for five pre-trained tokenizers (Table 1). XGLM consistently has the highest morphological alignment as measured by precision. The other tokenizers' rankings change depending on the different conditions. Measured with recall, Llama2 has the best recall. This is likely due to pervasive oversegmentations. Because of the variable rankings across different metrics and conditions, we determine the optimal default evaluation settings not by maximizing alignments scores, but by determining which is most predictive of language model performance.

## 4. Correlation with Language Model Performance

We replicate and expand the analysis in Arnett & Bergen (2025). We take reported model performance scores on a variety of downstream tasks in a range of languages. We test whether there is a correlation between morphological alignment and downstream performance. This serves two purposes. First, we can determine which settings are most predictive of model performance. This could inform choice of settings. Second, we replicate the analysis in Arnett & Bergen (2025), but with the inclusion of many more languages and additional models and tasks.

**Method.** We use reported model task performance results from Arnett & Bergen (2025). This includes tasks such as XCOPA (Ponti et al., 2020), XNLI (Conneau et al., 2018), and SIB-200 (Adelani et al., 2024). Scores come from Llama2 8B (Touvron et al., 2023), BLOOM (560M, 1.1B, 3B, 7.1B; Le Scao et al., 2023), and XGLM 7.5B (Lin et al., 2021). We add MultiBLiMP (Jumelet et al., 2025), which tests models' subject-verb agreement performance. We use the results for Llama3 (8B and 70B; Grattafiori et al., 2024) and Gemma3 (4B, 12B, 27B; Team et al., 2025), as reported in the MultiBLiMP paper. The inclusion of MultiBLiMP means we have performance results for all languages in our sample, since MultiBLiMP is also derived from UD. Following the previous study, we use the estimated training data proportions from Hayase et al. (2024), as the model

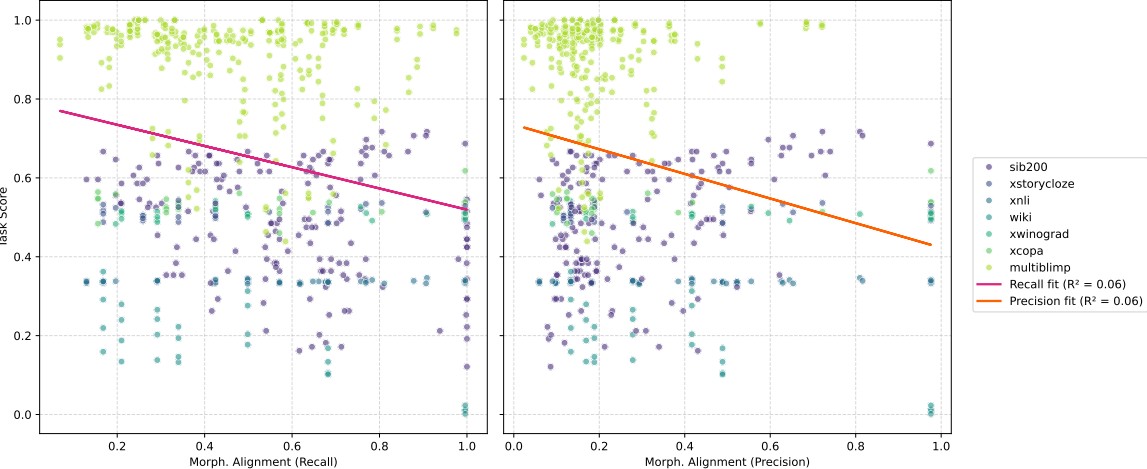

*Figure 2.* Correlation between model performance on different tasks (color-coded) for recall (left) and precision (right).

developers do not release that information about the pre-training data.

We test the correlation using linear mixed effects models. As it is known that model size, in parameters, and proportion of the training data in each language impact performance (Kaplan et al., 2020 and Bagheri Nezhad & Agrawal, 2024; Li et al., 2024, respectively), we include these factors as fixed effects. We included benchmark task as a random intercept, as the tasks have different levels of difficulty. We test whether morphological alignment explains additional variance above and beyond these factors using an ANOVA. We also use a simple linear regression to test how much variance morphological alignment explains in the model performance scores.

**Results.** We find that the fixed effects, number of parameters and proportion of training data in each languages, explains significantly more variance than the intercept ($\chi^2(2) = 25.67$, $p < 0.001$). Morphological alignment, as measured with recall, explains additional variance above and beyond these factors ($\chi^2(1) = 391.42$, $p < 0.001$); however, precision does not ($\chi^2(1) = -6.99$, $p = 1$).

Next, we report the amount of variance explained by morphological alignment. We find that the full linear mixed effects model only explains a small fraction of the variance (recall $R^2 = 0.024$, precision $R^2 = 0.005$). We also plot the relationship between both metrics of morphological alignment and model performance in Figure 2.

In addition to being a very small effect, the correlation between morphological alignment and model performance is *negative*. This is consistent with the findings in Arnett & Bergen (2025), and challenges claims that morpholog-

ically aligned tokenization can contribute to better model performance.

Comparing across condition, we find that the condition which frequency-scales scores and does not include one-token words has slightly more explanatory power for model performance, though we note this difference is numeric and the amount of variance is still quite small. All of the conditions still show small negative correlations with model performance. Therefore, we consider this setting to be an appropriate set of default scoring parameters. We report correlations for each condition in Appendix D.

## 5. Discussion

**Optimal Settings.** We tested a variety of parameters in our scoring function, and frequency scaling the scores and leaving out one-token words leads to slightly better prediction of model performance. This suggests that including frequency information does improve predictive power of our morphological alignment metrics. Our frequency metrics came only from the treebanks we used to create our datasets, meaning for some languages the sample was very small. Additionally, many treebanks are created with data from one source, e.g. news articles. In the future, word frequency could be calculated using larger corpora from a wider range of domains. Another possible change would be to use lemma frequency instead of wordform frequency. Particularly for agglutinative languages, e.g. Turkish, individual wordforms tend to be lower frequency. Any given verb, for example, can have thousands of different forms (Hakkani-Tür et al., 2002). Especially if we aim for our morphological alignment metric to capture how often a tokenizer encodes a word with semantically meaningful tokens,

like stems, measuring frequency by the lemma may improve predictive power of our morphological alignment score.

**The Relevance of Morphological Alignment.** Our results show that our version of morphological alignment score explains relatively little variance in model performance, even after taking into account model size and training data proportion. Given large amount of evidence in support of and against the claim that morphological tokenization helps model performance, these results should not be taken as conclusive. But, maybe it suggests that the relationship should be measured differently. Perhaps, taken in isolation, morphological alignment is not sufficient to classify tokenization as optimal. This seems plausible, given that we saw such a strong tradeoff between compression and morphological alignment, when we use accuracy as a metric. Combining morphological alignment with other intrinsic tokenizer evaluation metrics, like compression or Rényi efficiency, could potentially be more informative.

**Future Work.** While morphological alignment is not predictive of model performance as it is measured here, we hope our datasets and evaluation metric can be used to better understand multilingual tokenization. There are aspects of our evaluation we do not discuss here. Our implementation offers the ability to retrieve morphological alignment score broken down by POS, for instance. Our evaluation framework is flexible to allow many fine-grained analyses, which may be of interest to the wider research community.

## 6. Conclusion

In this paper, we develop and expanded and updated evaluation for tokenizer morphological alignment for over 70 languages. We test the impact of several design decisions in the scoring function, and find that the way that alignment is calculated leads to different morphological alignment scores and relative rankings between tokenizers. We also test whether morphological alignment is predictive of model performance, which is predicted by previous work. We find, however, that morphological alignment offers only a small negative correlation. This is consistent with the claim that morphologically aligned tokenization does not positively impact model performance. We release our evaluation framework and our datasets to support more work in this area to better understand what features of tokenizers are associated with better performance.

## Limitations

While we significantly expand language coverage of this type of tokenizer evaluation, our language sample is far from comprehensive. Additionally, European languages are over-represented in our sample. This is a result of systemic over-representations in the field and in resources like Universal Dependencies. Other resources like UniMorph could be used to improve language coverage, but UniMorph does not provide sentential context, so additional work would be needed to fully integrate UniMorph data into the framework we developed. We hope that as language coverage continually expands and diversifies, it will be easier to represent a more diverse sample of languages.

As with the original implementation of MorphScore, the operationalization of morphological boundaries is coarse. We aim mainly to capture the most clear-cut cases. This means that we mostly cover inflectional morphology and items that appear as single orthographic words. This undoubtedly misses many informative cases.

In this paper, we use only a small number of tasks to represent model performance. We used evaluations which were available for a wide variety of languages, but such evaluations are limited and generally do not represent most of the languages in our sample. For example XCOPA (Ponti et al., 2020) represents 11 languages and XNLI (Conneau et al., 2018) represents 15 languages. Many of these are high-resource European languages like English, Italian, German, and French or widely spoken languages that have been historically underrepresented in NLP like Swahili and Urdu.

Our focus is on large, autoregressive LMs, which allows for cleaner comparisons but excludes encoder models or those trained with masked language modeling. Our sample of models was not very large, because many models do not provide critical information about their training data. BLOOM and XGLM are the only models which report their training data proportions. We were able to expand our sample of models because of the work by Hayase et al. (2024) estimating training data proportions by language for closed-data models. We also chose to exclude instruction-tuned models, because similarly information about fine-tuning data proportions by language was not available. Furthermore,it was not clear about how to calculate proportion of training data, taking into account pre-training data proportions and fine-tuning data proportions.

## Impact Statement

Our work aims to understand tokenization quality, which is an issue that disproportionately affects low-resource languages (Petrov et al., 2023; Ahia et al., 2023). We hope that our work positively contributes towards understanding relevant features of tokenization in a multilingual context and helps improve equity in language technology performance across languages.

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

# A. Language Sample

Table 2 reports the number of items for each language after filtering and the UD treebank used to create the dataset for that language. The versions of each treebank that are used are reported in the Hugging Face README: https://huggingface.co/datasets/catherinearnett/morphscore.

*Table 2.* List of languages, UD sources, and number of items after filtering

| Language | ISO 639-3 | ISO 15924 | Num. Items | Data Source |
|---|---|---|---|---|
| Afrikaans | afr | latn | 1397 | UD_Afrikaans-AfriBooms (Augustinus et al., 2016) |
| Albanian | sqi | latn | 366 | UD_Albanian-STAF (Talamo, 2025; Kote et al., 2024) |
| Armenian | hye | armn | 5441 | UD_Armenian-ArmTDP (Yavrumyan & Anna, 2020) |
| Azerbaijani | aze | latn | 220 | UD_Azerbaijani-TueCL (Eslami & Çağrı Çöltekin, 2024) |
| Basque | eus | latn | 12089 | UD_Basque-BDT (Aranzabe et al., 2015) |
| Belarusian | bel | cyrl | 9935 | UD_Belarusian-HSE (Shishkina & Lyashevskaya, 2021) |
| Bhojpuri | bho | deva | 177 | UD_Bhojpuri-BHTB (Ojha & Zeman, 2020) |
| Breton | bre | latn | 233 | UD_Breton-KEB (Tyers & Ravishankar, 2018) |
| Bulgarian | bul | cyrl | 5443 | UD_Bulgarian-BTB (Simov et al., 2004) |
| Buriat | bur | cyrl | 1983 | UD_Buryat-BDT (Badmaeva & Tyers, 2017) |
| Catalan | cat | latn | 1230 | UD_Catalan-AnCora (Taulé et al., 2008) |
| Croatian | hrv | latn | 7749 | UD_Croatian-SET (Agić & Ljubešić, 2015) |
| Czech | ces | latn | 15059 | UD_Czech-CAC (Hladká et al., 2008) (Bejček et al., 2022) |
| Danish | dan | latn | 6680 | UD_Danish-DDT (Johannsen et al., 2015) |
| Dutch | nld | latn | 3606 | UD_Dutch-Alpino (Van der Beek et al., 2002) |
| English | eng | latn | 3688 | UD_English-EWT (Silveira et al., 2014) |
| Erzya | myv | cyrl | 2309 | UD_Erzya-JR (Rueter & Tyers, 2018) |
| Estonian | est | latn | 19261 | UD_Estonian-EDT (Muischnek et al., 2014) |
| Finnish | fin | latn | 10172 | UD_Finnish-TDT (Haverinen et al., 2014) (Pyysalo et al., 2015) |
| French | fra | latn | 6082 | UD_French-GSD (Guillaume et al., 2019) |
| Galician | glg | latn | 2879 | UD_Galician-CTG (Guinovart, 2017) |
| Georgian | kat | geor | 2535 | UD_Georgian-GLC (Lobzhanidze, 2022) |
| German | deu | latn | 31281 | UD_German-HDT (Borges Völker et al., 2019) |
| Greek | ell | grek | 691 | UD_Greek-GDT (Prokopidis et al., 2005) (Prokopidis & Papageorgiou, 2017) |
| Hebrew | heb | hebr | 4641 | UD_Hebrew-HTB (Tsarfaty, 2013) (McDonald et al., 2013b) |
| Hindi | hin | deva | 1301 | UD_Hindi-HDTB (Palmer et al., 2009; Bhat et al., 2017) |
| Hungarian | hun | latn | 6350 | UD_Hungarian-Szeged (Vincze et al., 2010) |
| Icelandic | isl | latn | 13155 | UD_Icelandic-IcePaHC (Arnardóttir et al., 2020; 2023) |
| Indonesian | ind | latn | 2785 | UD_Indonesian-GSD (Larasati et al., 2011) (McDonald et al., 2013a) |
| Irish | gle | latn | 2576 | UD_Irish-IDT (Lynn, 2016) |
| Kazakh | kaz | cyrl | 2442 | UD_Kazakh-KTB (Tyers & Washington, 2015) (Makazhanov et al., 2015) |
| Kirghiz | kir | cyrl | 4221 | UD_Kyrgyz-KTMU (İbrahim Benli, 2023) |
| Komi-Zyrian | kpv | cyrl | 1038 | UD_Komi_Zyrian-Lattice (Partanen et al., 2018) |
| Korean | kor | hang | 316 | UD_Korean-Kaist (Chun et al., 2018) |
| Latvian | lav | latn | 8332 | UD_Latvian-LVTB (Pretkalniņa et al., 2018) |
| Lithuanian | lit | latn | 667 | UD_Lithuanian-ALKSNIS (Bielinskienė et al., 2016) |
| Macedonian | mkd | cyrl | 153 | UD_Macedonian-MTB (Sazdov, 2012) |

*Table 2.* List of languages, UD sources, and number of items after filtering (continued)

| Language | ISO 639-3 | ISO 15924 | Num. Items | Data Source | |
|---|---|---|---|---|---|
| Malayalam | mal | mlym | 131 | UD_Malayalam-UFAL | (Sharma et al., 2021) |
| Manx | glv | latn | 224 | UD_Manx-Cadhan | (Scannell, 2020) |
| Marathi | mar | deva | 171 | UD_Marathi-UFAL | (Ravishankar, 2017) |
| Moksha | mdf | cyrl | 615 | UD_Moksha-JR | (Rueter, 2018) |
| Northern Sami | sme | latn | 664 | UD_North_Sami-Giella | (Sheyanova & Tyers, 2017) |
| Norwegian | nob | latn | 13017 | UD_Norwegian-Bokmaal | (Solberg et al., 2014) |
| Occitan | oci | latn | 878 | UD_Occitan-TTB | (Miletic et al., 2020) |
| Pashto | pus | arab | 155 | UD_Pashto-Sikaram | (Faryad & Zeman, 2024) |
| Persian | fas | arab | 11859 | UD_Persian-PerDT | (Etezadi et al., 2022) |
| Polish | pol | latn | 10886 | UD_Polish-PDB | (Wróblewska, 2018) |
| Portuguese | por | latn | 4559 | UD_Portuguese-CINTIL | (Branco et al., 2022) |
| Romanian | ron | latn | 10129 | UD_Romanian-RRT | (Irimia & Mititelu, 2015) |
| Russian | rus | cyrl | 21569 | UD_Russian-SynTagRus | (Droganova et al., 2018) |
| Sanskrit | san | deva | 16184 | UD_Sanskrit-Vedic | (Hellwig et al., 2020; 2023) |
| Scottish Gaelic | gla | latn | 1004 | UD_Scottish_Gaelic-ARCOSG | (Batchelor, 2019) |
| Serbian | srp | latn | 3874 | UD_Serbian-SET | (Samardžić & Ljubešić, 2024) |
| Sindhi | snd | arab | 3874 | UD_Sindhi-Isra | (Rahman et al., 2024) |
| Sinhala | sin | sinh | 196 | UD_Sinhala-STB | (Liyanage et al., 2023) |
| Slovak | slk | latn | 3590 | UD_Slovak-SNK | (Zeman, 2017) |
| Slovenian | slv | latn | 11383 | UD_Slovenian-SSJ | (Dobrovoljc et al., 2017) |
| | | | | | (Dobrovoljc & Ljubešić, 2022) |
| Spanish | spa | latn | 6658 | UD_Spanish-AnCora | (Taulé et al., 2008) |
| Swedish | swe | latn | 6223 | UD_Swedish-LinES | (Ahrenberg, 2007) |
| Tamil | tam | taml | 1179 | UD_Tamil-TTB | (Ramasamy & Žabokrtský, 2012) |
| Tatar | tat | cyrl | 627 | UD_Tatar-NMCTT | (Taguchi, 2024) |
| Turkish | tur | latn | 30076 | UD_Turkish-Kenet | (Kuzgun et al., 2021) |
| Uighur | uig | arab | 3073 | UD_Uyghur-UDT | (Eli et al., 2024) |
| Ukrainian | ukr | cyrl | 4182 | UD_Ukrainian-IU | (Kotsyba et al., 2024) |
| Upper Sorbian | hsb | latn | 867 | UD_Upper_Sorbian-UFAL | (Zeman & Nedoluzhko, 2024) |
| Urdu | urd | arab | 981 | UD_Urdu-UDTB | (Bhat et al., 2017) |
| Uzbek | uzb | latn | 1867 | UD_Uzbek-UT | (Akhundjanova & Talamo, 2025) |
| Veps | vep | latn | 159 | UD_Veps-VWT | (Laan, 2024) |
| Welsh | cym | latn | 757 | UD_Welsh-CCG | (Heinecke & Tyers, 2019) |
| Wolof | wol | latn | 1355 | UD_Wolof-WTB | (Dione, 2024) |
| Yakut | sah | cyrl | 250 | UD_Yakut-YKTDT | (Merzhevich & Gerardi, 2022) |

# B. MorphScore by Language

*Table 3.* Precision (± standard deviation) for each language for all tokenizers tested.

| Language | BLOOM | XGLM | Gemma3 | Llama2 | Llama3 |
|---|---|---|---|---|---|
| Afrikaans | 0.11 ± 0.00 | 0.44 ± 0.00 | 0.12 ± 0.00 | 0.09 ± 0.00 | 0.10 ± 0.00 |
| Albanian | 0.12 ± 0.00 | 0.31 ± 0.00 | 0.12 ± 0.00 | 0.14 ± 0.00 | 0.14 ± 0.00 |
| Armenian | 0.12 ± 0.00 | 0.34 ± 0.00 | 0.19 ± 0.00 | 0.11 ± 0.00 | 0.07 ± 0.00 |
| Azerbaijani | 0.22 ± 0.00 | 0.38 ± 0.00 | 0.20 ± 0.00 | 0.16 ± 0.00 | 0.17 ± 0.00 |
| Basque | 0.12 ± 0.00 | 0.26 ± 0.00 | 0.12 ± 0.00 | 0.12 ± 0.00 | 0.11 ± 0.00 |
| Belarusian | 0.10 ± 0.00 | 0.37 ± 0.00 | 0.08 ± 0.00 | 0.10 ± 0.00 | 0.11 ± 0.00 |
| Bhojpuri | 0.21 ± 0.00 | 0.40 ± 0.00 | 0.33 ± 0.00 | 0.19 ± 0.00 | 0.21 ± 0.00 |
| Breton | 0.36 ± 0.00 | 0.35 ± 0.00 | 0.30 ± 0.00 | 0.21 ± 0.00 | 0.25 ± 0.00 |
| Bulgarian | 0.10 ± 0.00 | 0.47 ± 0.00 | 0.09 ± 0.00 | 0.13 ± 0.00 | 0.11 ± 0.00 |
| Buriat | 0.15 ± 0.00 | 0.22 ± 0.00 | 0.17 ± 0.00 | 0.15 ± 0.00 | 0.14 ± 0.00 |
| Catalan | 0.43 ± 0.00 | 0.74 ± 0.00 | 0.38 ± 0.00 | 0.44 ± 0.00 | 0.37 ± 0.00 |
| Croatian | 0.11 ± 0.00 | 0.55 ± 0.00 | 0.13 ± 0.00 | 0.13 ± 0.00 | 0.13 ± 0.00 |
| Czech | 0.19 ± 0.00 | 0.43 ± 0.00 | 0.14 ± 0.00 | 0.17 ± 0.00 | 0.14 ± 0.00 |
| Danish | 0.13 ± 0.00 | 0.50 ± 0.00 | 0.16 ± 0.00 | 0.11 ± 0.00 | 0.12 ± 0.00 |
| Dutch | 0.10 ± 0.00 | 0.56 ± 0.00 | 0.14 ± 0.00 | 0.18 ± 0.00 | 0.12 ± 0.00 |
| English | 0.42 ± 0.00 | 0.81 ± 0.00 | 0.72 ± 0.00 | 0.72 ± 0.00 | 0.58 ± 0.00 |
| Erzya | 0.14 ± 0.00 | 0.16 ± 0.00 | 0.17 ± 0.00 | 0.16 ± 0.00 | 0.11 ± 0.00 |
| Estonian | 0.10 ± 0.00 | 0.31 ± 0.00 | 0.11 ± 0.00 | 0.11 ± 0.00 | 0.11 ± 0.00 |
| Finnish | 0.12 ± 0.00 | 0.43 ± 0.00 | 0.12 ± 0.00 | 0.13 ± 0.00 | 0.11 ± 0.00 |
| French | 0.28 ± 0.00 | 0.65 ± 0.00 | 0.28 ± 0.00 | 0.36 ± 0.00 | 0.20 ± 0.00 |
| Galician | 0.35 ± 0.00 | 0.70 ± 0.00 | 0.33 ± 0.00 | 0.34 ± 0.00 | 0.29 ± 0.00 |
| Georgian | 0.09 ± 0.00 | 0.23 ± 0.00 | 0.06 ± 0.00 | 0.11 ± 0.00 | 0.06 ± 0.00 |
| German | 0.06 ± 0.00 | 0.63 ± 0.00 | 0.15 ± 0.00 | 0.36 ± 0.00 | 0.07 ± 0.00 |
| Greek | 0.52 ± 0.00 | 0.82 ± 0.00 | 0.68 ± 0.00 | 0.31 ± 0.00 | 0.68 ± 0.00 |
| Hebrew | 0.29 ± 0.00 | 0.40 ± 0.00 | 0.27 ± 0.00 | 0.24 ± 0.00 | 0.25 ± 0.00 |
| Hindi | 0.49 ± 0.00 | 0.65 ± 0.00 | 0.36 ± 0.00 | 0.16 ± 0.00 | 0.18 ± 0.00 |
| Hungarian | 0.19 ± 0.00 | 0.42 ± 0.00 | 0.19 ± 0.00 | 0.18 ± 0.00 | 0.17 ± 0.00 |
| Icelandic | 0.14 ± 0.00 | 0.30 ± 0.00 | 0.16 ± 0.00 | 0.16 ± 0.00 | 0.16 ± 0.00 |
| Indonesian | 0.19 ± 0.00 | 0.72 ± 0.00 | 0.20 ± 0.00 | 0.19 ± 0.00 | 0.19 ± 0.00 |
| Irish | 0.31 ± 0.00 | 0.37 ± 0.00 | 0.32 ± 0.00 | 0.25 ± 0.00 | 0.31 ± 0.00 |
| Kazakh | 0.17 ± 0.00 | 0.46 ± 0.00 | 0.18 ± 0.00 | 0.16 ± 0.00 | 0.15 ± 0.00 |
| Kirghiz | 0.16 ± 0.00 | 0.30 ± 0.00 | 0.16 ± 0.00 | 0.15 ± 0.00 | 0.15 ± 0.00 |
| Komi | 0.16 ± 0.00 | 0.20 ± 0.00 | 0.17 ± 0.00 | 0.16 ± 0.00 | 0.17 ± 0.00 |
| Korean | 0.29 ± 0.00 | 0.73 ± 0.00 | 0.66 ± 0.00 | 0.18 ± 0.00 | 0.60 ± 0.00 |
| Latvian | 0.02 ± 0.00 | 0.39 ± 0.00 | 0.02 ± 0.00 | 0.04 ± 0.00 | 0.04 ± 0.00 |
| Lithuanian | 0.16 ± 0.00 | 0.49 ± 0.00 | 0.04 ± 0.00 | 0.16 ± 0.00 | 0.12 ± 0.00 |
| Macedonian | 0.17 ± 0.00 | 0.42 ± 0.00 | 0.17 ± 0.00 | 0.17 ± 0.00 | 0.09 ± 0.00 |
| Malayalam | 0.10 ± 0.00 | 0.23 ± 0.00 | 0.15 ± 0.00 | 0.08 ± 0.00 | 0.05 ± 0.00 |
| Mandarin Chinese | 0.98 ± 0.02 | 0.09 ± 0.00 | 0.94 ± 0.02 | 0.31 ± 0.01 | 0.90 ± 0.02 |
| Manx | 0.20 ± 0.00 | 0.15 ± 0.00 | 0.20 ± 0.00 | 0.19 ± 0.00 | 0.18 ± 0.00 |
| Marathi | 0.24 ± 0.00 | 0.28 ± 0.00 | 0.09 ± 0.00 | 0.14 ± 0.00 | 0.23 ± 0.00 |
| Moksha | 0.20 ± 0.00 | 0.22 ± 0.00 | 0.20 ± 0.00 | 0.20 ± 0.00 | 0.17 ± 0.00 |
| Norwegian | 0.20 ± 0.00 | 0.59 ± 0.00 | 0.21 ± 0.00 | 0.19 ± 0.00 | 0.17 ± 0.00 |
| Occitan | 0.32 ± 0.00 | 0.48 ± 0.00 | 0.34 ± 0.00 | 0.33 ± 0.00 | 0.33 ± 0.00 |
| Pashto | 0.15 ± 0.00 | 0.44 ± 0.00 | 0.17 ± 0.00 | 0.16 ± 0.00 | 0.16 ± 0.00 |
| Persian | 0.25 ± 0.00 | 0.55 ± 0.00 | 0.23 ± 0.00 | 0.16 ± 0.00 | 0.20 ± 0.00 |

*Table 4.* Precision (± standard deviation) for each language for all tokenizers tested (continued).

| Language | BLOOM | XGLM | Gemma3 | Llama2 | Llama3 |
|---|---|---|---|---|---|
| Polish | 0.15 ± 0.00 | 0.41 ± 0.00 | 0.15 ± 0.00 | 0.17 ± 0.00 | 0.14 ± 0.00 |
| Portuguese | 0.17 ± 0.00 | 0.60 ± 0.00 | 0.15 ± 0.00 | 0.15 ± 0.00 | 0.13 ± 0.00 |
| Romanian | 0.19 ± 0.00 | 0.50 ± 0.00 | 0.19 ± 0.00 | 0.20 ± 0.00 | 0.20 ± 0.00 |
| Russian | 0.13 ± 0.00 | 0.56 ± 0.00 | 0.12 ± 0.00 | 0.17 ± 0.00 | 0.17 ± 0.00 |
| Sami | 0.07 ± 0.00 | 0.14 ± 0.00 | 0.09 ± 0.00 | 0.08 ± 0.00 | 0.08 ± 0.00 |
| Sanskrit | 0.15 ± 0.00 | 0.14 ± 0.00 | 0.16 ± 0.00 | 0.15 ± 0.00 | 0.13 ± 0.00 |
| Scottish Gaelic | 0.33 ± 0.00 | 0.49 ± 0.00 | 0.44 ± 0.00 | 0.28 ± 0.00 | 0.34 ± 0.00 |
| Serbian | 0.11 ± 0.00 | 0.61 ± 0.00 | 0.13 ± 0.00 | 0.14 ± 0.00 | 0.13 ± 0.00 |
| Sindhi | 0.22 ± 0.00 | 0.35 ± 0.00 | 0.23 ± 0.00 | 0.17 ± 0.00 | 0.21 ± 0.00 |
| Sinhala | 0.09 ± 0.00 | 0.43 ± 0.00 | 0.15 ± 0.00 | 0.08 ± 0.00 | 0.09 ± 0.00 |
| Slovak | 0.21 ± 0.00 | 0.45 ± 0.00 | 0.18 ± 0.00 | 0.19 ± 0.00 | 0.17 ± 0.00 |
| Slovenian | 0.12 ± 0.00 | 0.47 ± 0.00 | 0.14 ± 0.00 | 0.15 ± 0.00 | 0.14 ± 0.00 |
| Spanish | 0.13 ± 0.00 | 0.63 ± 0.00 | 0.15 ± 0.00 | 0.21 ± 0.00 | 0.10 ± 0.00 |
| Swedish | 0.14 ± 0.00 | 0.48 ± 0.00 | 0.24 ± 0.00 | 0.22 ± 0.00 | 0.14 ± 0.00 |
| Tamil | 0.12 ± 0.00 | 0.42 ± 0.00 | 0.14 ± 0.00 | 0.10 ± 0.00 | 0.08 ± 0.00 |
| Tatar | 0.15 ± 0.00 | 0.23 ± 0.00 | 0.18 ± 0.00 | 0.15 ± 0.00 | 0.14 ± 0.00 |
| Turkish | 0.18 ± 0.00 | 0.38 ± 0.00 | 0.20 ± 0.00 | 0.14 ± 0.00 | 0.18 ± 0.00 |
| Uighur | 0.16 ± 0.00 | 0.21 ± 0.00 | 0.18 ± 0.00 | 0.13 ± 0.00 | 0.15 ± 0.00 |
| Ukrainian | 0.12 ± 0.00 | 0.42 ± 0.00 | 0.05 ± 0.00 | 0.06 ± 0.00 | 0.12 ± 0.00 |
| Upper Sorbian | 0.16 ± 0.00 | 0.22 ± 0.00 | 0.14 ± 0.00 | 0.16 ± 0.00 | 0.15 ± 0.00 |
| Urdu | 0.29 ± 0.00 | 0.50 ± 0.00 | 0.25 ± 0.00 | 0.14 ± 0.00 | 0.15 ± 0.00 |
| Uzbek | 0.19 ± 0.00 | 0.25 ± 0.00 | 0.20 ± 0.00 | 0.16 ± 0.00 | 0.18 ± 0.00 |
| Veps | 0.12 ± 0.00 | 0.19 ± 0.00 | 0.16 ± 0.00 | 0.17 ± 0.00 | 0.15 ± 0.00 |
| Welsh | 0.43 ± 0.00 | 0.62 ± 0.00 | 0.49 ± 0.00 | 0.38 ± 0.00 | 0.43 ± 0.00 |
| Wolof | 0.18 ± 0.00 | 0.17 ± 0.00 | 0.20 ± 0.00 | 0.19 ± 0.00 | 0.21 ± 0.00 |
| Yakut | 0.16 ± 0.00 | 0.18 ± 0.00 | 0.15 ± 0.00 | 0.14 ± 0.00 | 0.17 ± 0.00 |

*Table 5.* Recall (± standard deviation) for each language for all tokenizers tested.

| Language | BLOOM | XGLM | Gemma3 | Llama2 | Llama3 |
|---|---|---|---|---|---|
| Afrikaans | 0.25 ± 0.00 | 0.64 ± 0.00 | 0.23 ± 0.00 | 0.21 ± 0.00 | 0.26 ± 0.00 |
| Albanian | 0.31 ± 0.00 | 0.52 ± 0.00 | 0.29 ± 0.00 | 0.41 ± 0.00 | 0.41 ± 0.00 |
| Armenian | 0.68 ± 0.00 | 0.65 ± 0.00 | 0.84 ± 0.00 | 1.00 ± 0.00 | 0.89 ± 0.00 |
| Azerbaijani | 0.64 ± 0.00 | 0.66 ± 0.00 | 0.55 ± 0.00 | 0.68 ± 0.00 | 0.56 ± 0.00 |
| Basque | 0.26 ± 0.00 | 0.55 ± 0.00 | 0.33 ± 0.00 | 0.41 ± 0.00 | 0.35 ± 0.00 |
| Belarusian | 0.43 ± 0.00 | 0.55 ± 0.00 | 0.20 ± 0.00 | 0.33 ± 0.00 | 0.36 ± 0.00 |
| Bhojpuri | 0.32 ± 0.00 | 0.61 ± 0.00 | 0.49 ± 0.00 | 1.00 ± 0.00 | 0.62 ± 0.00 |
| Breton | 0.60 ± 0.00 | 0.58 ± 0.00 | 0.57 ± 0.00 | 0.53 ± 0.00 | 0.49 ± 0.00 |
| Bulgarian | 0.42 ± 0.00 | 0.66 ± 0.00 | 0.25 ± 0.00 | 0.43 ± 0.00 | 0.38 ± 0.00 |
| Buriat | 0.76 ± 0.00 | 0.55 ± 0.00 | 0.67 ± 0.00 | 0.73 ± 0.00 | 0.69 ± 0.00 |
| Catalan | 0.48 ± 0.00 | 0.87 ± 0.00 | 0.43 ± 0.00 | 0.53 ± 0.00 | 0.45 ± 0.00 |
| Croatian | 0.29 ± 0.00 | 0.71 ± 0.00 | 0.29 ± 0.00 | 0.36 ± 0.00 | 0.37 ± 0.00 |
| Czech | 0.57 ± 0.00 | 0.63 ± 0.00 | 0.29 ± 0.00 | 0.41 ± 0.00 | 0.31 ± 0.00 |
| Danish | 0.27 ± 0.00 | 0.65 ± 0.00 | 0.25 ± 0.00 | 0.24 ± 0.00 | 0.24 ± 0.00 |
| Dutch | 0.21 ± 0.00 | 0.69 ± 0.00 | 0.23 ± 0.00 | 0.30 ± 0.00 | 0.25 ± 0.00 |
| English | 0.50 ± 0.00 | 0.91 ± 0.00 | 0.75 ± 0.00 | 0.76 ± 0.00 | 0.64 ± 0.00 |
| Erzya | 0.60 ± 0.00 | 0.48 ± 0.00 | 0.54 ± 0.00 | 0.63 ± 0.00 | 0.38 ± 0.00 |
| Estonian | 0.31 ± 0.00 | 0.48 ± 0.00 | 0.30 ± 0.00 | 0.38 ± 0.00 | 0.35 ± 0.00 |

*Table 6.* Recall (± standard deviation) for each language for all tokenizers tested (continued).

| Language | BLOOM | XGLM | Gemma3 | Llama2 | Llama3 |
|---|---|---|---|---|---|
| Finnish | 0.33 ± 0.00 | 0.67 ± 0.00 | 0.29 ± 0.00 | 0.39 ± 0.00 | 0.32 ± 0.00 |
| French | 0.29 ± 0.00 | 0.78 ± 0.00 | 0.31 ± 0.00 | 0.39 ± 0.00 | 0.23 ± 0.00 |
| Galician | 0.43 ± 0.00 | 0.82 ± 0.00 | 0.43 ± 0.00 | 0.46 ± 0.00 | 0.40 ± 0.00 |
| Georgian | 0.66 ± 0.00 | 0.42 ± 0.00 | 0.21 ± 0.00 | 1.00 ± 0.00 | 0.98 ± 0.00 |
| German | 0.13 ± 0.00 | 0.77 ± 0.00 | 0.23 ± 0.00 | 0.44 ± 0.00 | 0.16 ± 0.00 |
| Greek | 0.68 ± 0.00 | 0.88 ± 0.00 | 0.70 ± 0.00 | 1.00 ± 0.00 | 0.70 ± 0.00 |
| Hebrew | 0.94 ± 0.00 | 0.72 ± 0.00 | 0.64 ± 0.00 | 1.00 ± 0.00 | 0.89 ± 0.00 |
| Hindi | 0.68 ± 0.00 | 0.91 ± 0.00 | 0.67 ± 0.00 | 1.00 ± 0.00 | 0.63 ± 0.00 |
| Hungarian | 0.67 ± 0.00 | 0.73 ± 0.00 | 0.58 ± 0.00 | 0.63 ± 0.00 | 0.64 ± 0.00 |
| Icelandic | 0.45 ± 0.00 | 0.54 ± 0.00 | 0.46 ± 0.00 | 0.55 ± 0.00 | 0.55 ± 0.00 |
| Indonesian | 0.34 ± 0.00 | 0.81 ± 0.00 | 0.37 ± 0.00 | 0.52 ± 0.00 | 0.42 ± 0.00 |
| Irish | 0.54 ± 0.00 | 0.53 ± 0.00 | 0.56 ± 0.00 | 0.59 ± 0.00 | 0.56 ± 0.00 |
| Kazakh | 0.75 ± 0.00 | 0.71 ± 0.00 | 0.58 ± 0.00 | 0.67 ± 0.00 | 0.62 ± 0.00 |
| Kirghiz | 0.81 ± 0.00 | 0.65 ± 0.00 | 0.55 ± 0.00 | 0.72 ± 0.00 | 0.68 ± 0.00 |
| Komi | 0.85 ± 0.00 | 0.69 ± 0.00 | 0.69 ± 0.00 | 0.83 ± 0.00 | 0.81 ± 0.00 |
| Korean | 1.00 ± 0.00 | 0.79 ± 0.00 | 0.99 ± 0.00 | 1.00 ± 0.00 | 0.98 ± 0.00 |
| Latvian | 0.07 ± 0.00 | 0.52 ± 0.00 | 0.07 ± 0.00 | 0.16 ± 0.00 | 0.16 ± 0.00 |
| Lithuanian | 0.66 ± 0.00 | 0.62 ± 0.00 | 0.13 ± 0.00 | 0.71 ± 0.00 | 0.67 ± 0.00 |
| Macedonian | 0.69 ± 0.00 | 0.70 ± 0.00 | 0.46 ± 0.00 | 0.58 ± 0.00 | 0.30 ± 0.00 |
| Malayalam | 0.29 ± 0.00 | 0.68 ± 0.00 | 0.59 ± 0.00 | 1.00 ± 0.00 | 0.90 ± 0.00 |
| Mandarin Chinese | 1.00 ± 0.02 | 0.20 ± 0.00 | 1.00 ± 0.02 | 1.00 ± 0.02 | 1.00 ± 0.02 |
| Manx | 0.62 ± 0.00 | 0.38 ± 0.00 | 0.55 ± 0.00 | 0.62 ± 0.00 | 0.56 ± 0.00 |
| Marathi | 0.54 ± 0.00 | 0.55 ± 0.00 | 0.18 ± 0.00 | 1.00 ± 0.00 | 0.87 ± 0.00 |
| Moksha | 0.79 ± 0.00 | 0.61 ± 0.00 | 0.64 ± 0.00 | 0.71 ± 0.00 | 0.59 ± 0.00 |
| Norwegian | 0.34 ± 0.00 | 0.71 ± 0.00 | 0.30 ± 0.00 | 0.32 ± 0.00 | 0.30 ± 0.00 |
| Occitan | 0.37 ± 0.00 | 0.64 ± 0.00 | 0.41 ± 0.00 | 0.43 ± 0.00 | 0.40 ± 0.00 |
| Pashto | 0.47 ± 0.00 | 0.70 ± 0.00 | 0.47 ± 0.00 | 0.98 ± 0.00 | 0.55 ± 0.00 |
| Persian | 0.68 ± 0.00 | 0.79 ± 0.00 | 0.58 ± 0.00 | 1.00 ± 0.00 | 0.55 ± 0.00 |
| Polish | 0.42 ± 0.00 | 0.58 ± 0.00 | 0.31 ± 0.00 | 0.40 ± 0.00 | 0.35 ± 0.00 |
| Portuguese | 0.21 ± 0.00 | 0.77 ± 0.00 | 0.23 ± 0.00 | 0.25 ± 0.00 | 0.18 ± 0.00 |
| Romanian | 0.44 ± 0.00 | 0.69 ± 0.00 | 0.40 ± 0.00 | 0.47 ± 0.00 | 0.49 ± 0.00 |
| Russian | 0.42 ± 0.00 | 0.76 ± 0.00 | 0.15 ± 0.00 | 0.25 ± 0.00 | 0.42 ± 0.00 |
| Sami | 0.25 ± 0.00 | 0.45 ± 0.00 | 0.28 ± 0.00 | 0.36 ± 0.00 | 0.32 ± 0.00 |
| Sanskrit | 0.49 ± 0.00 | 0.36 ± 0.00 | 0.48 ± 0.00 | 0.56 ± 0.00 | 0.58 ± 0.00 |
| Scottish Gaelic | 0.49 ± 0.00 | 0.68 ± 0.00 | 0.65 ± 0.00 | 0.57 ± 0.00 | 0.54 ± 0.00 |
| Serbian | 0.29 ± 0.00 | 0.76 ± 0.00 | 0.30 ± 0.00 | 0.38 ± 0.00 | 0.36 ± 0.00 |
| Sindhi | 0.77 ± 0.00 | 0.64 ± 0.00 | 0.66 ± 0.00 | 1.00 ± 0.00 | 0.79 ± 0.00 |
| Sinhala | 1.00 ± 0.00 | 0.62 ± 0.00 | 0.53 ± 0.00 | 1.00 ± 0.00 | 1.00 ± 0.00 |
| Slovak | 0.57 ± 0.00 | 0.63 ± 0.00 | 0.35 ± 0.00 | 0.43 ± 0.00 | 0.37 ± 0.00 |
| Slovenian | 0.34 ± 0.00 | 0.64 ± 0.00 | 0.32 ± 0.00 | 0.41 ± 0.00 | 0.39 ± 0.00 |
| Spanish | 0.17 ± 0.00 | 0.77 ± 0.00 | 0.19 ± 0.00 | 0.27 ± 0.00 | 0.14 ± 0.00 |
| Swedish | 0.27 ± 0.00 | 0.63 ± 0.00 | 0.33 ± 0.00 | 0.38 ± 0.00 | 0.26 ± 0.00 |
| Tamil | 0.16 ± 0.00 | 0.68 ± 0.00 | 0.33 ± 0.00 | 1.00 ± 0.00 | 0.92 ± 0.00 |
| Tatar | 0.73 ± 0.00 | 0.65 ± 0.00 | 0.60 ± 0.00 | 0.69 ± 0.00 | 0.67 ± 0.00 |

*Table 7.* Recall (± standard deviation) for each language for all tokenizers tested (continued).

| Language | BLOOM | XGLM | Gemma3 | Llama2 | Llama3 |
|---|---|---|---|---|---|
| Turkish | 0.57 ± 0.00 | 0.65 ± 0.00 | 0.51 ± 0.00 | 0.56 ± 0.00 | 0.48 ± 0.00 |
| Uighur | 0.73 ± 0.00 | 0.71 ± 0.00 | 0.67 ± 0.00 | 1.00 ± 0.00 | 0.79 ± 0.00 |
| Ukrainian | 0.54 ± 0.00 | 0.66 ± 0.00 | 0.13 ± 0.00 | 0.20 ± 0.00 | 0.39 ± 0.00 |
| Upper Sorbian | 0.47 ± 0.00 | 0.38 ± 0.00 | 0.34 ± 0.00 | 0.42 ± 0.00 | 0.40 ± 0.00 |
| Urdu | 0.62 ± 0.00 | 0.81 ± 0.00 | 0.72 ± 0.00 | 1.00 ± 0.00 | 0.69 ± 0.00 |
| Uzbek | 0.57 ± 0.00 | 0.56 ± 0.00 | 0.53 ± 0.00 | 0.54 ± 0.00 | 0.57 ± 0.00 |
| Veps | 0.29 ± 0.00 | 0.38 ± 0.00 | 0.36 ± 0.00 | 0.48 ± 0.00 | 0.39 ± 0.00 |
| Welsh | 0.64 ± 0.00 | 0.72 ± 0.00 | 0.66 ± 0.00 | 0.77 ± 0.00 | 0.61 ± 0.00 |
| Wolof | 0.41 ± 0.00 | 0.35 ± 0.00 | 0.49 ± 0.00 | 0.56 ± 0.00 | 0.57 ± 0.00 |
| Yakut | 0.79 ± 0.00 | 0.51 ± 0.00 | 0.58 ± 0.00 | 0.71 ± 0.00 | 0.75 ± 0.00 |

## C. Full Statistical Results

Tables 8 and 9 report the results of the linear mixed effects models described in Section 4.

*Table 8.* Precision

| Variable | Coef. | Std. Err. | z | p-value |
|---|---|---|---|---|
| Intercept | 0.169 | 0.030 | 5.541 | 0.000 |
| **Frequency Scaling** | **0.102** | **0.008** | **13.565** | **0.000** |
| **Single-Token** | **-0.045** | **0.008** | **-5.929** | **0.000** |
| **Frequency Scaling × Single-Token** | **-0.087** | **0.011** | **-8.073** | **0.000** |
| Group Var | 0.005 | 0.025 | | |

*Table 9.* Recall

| Variable | Coef. | Std. Err. | z | p-value |
|---|---|---|---|---|
| Intercept | 0.476 | 0.042 | 11.336 | 0.000 |
| **Frequency Scaling** | **0.065** | **0.013** | **5.000** | **0.000** |
| **Single-Token** | **-0.036** | **0.013** | **-2.787** | **0.005** |
| **Frequency Scaling × Single-Token** | **-0.064** | **0.018** | **-3.459** | **0.001** |
| Group Var | 0.008 | 0.027 | | |

## D. Correlation with Model Performance in All Conditions

Figures 3 and 4 show the correlation between task performance by condition. The `True_True` condition indicates that scores were scaled by word frequency and single-token words were excluded. The `True_False` condition indicates that scores were scaled by word frequency and single-token words were included. The `False_True` condition indicates that scores were not scaled by word frequency and single-token words were excluded. The `False_False` condition indicates that scores were not scaled by word frequency and single-token words were included.

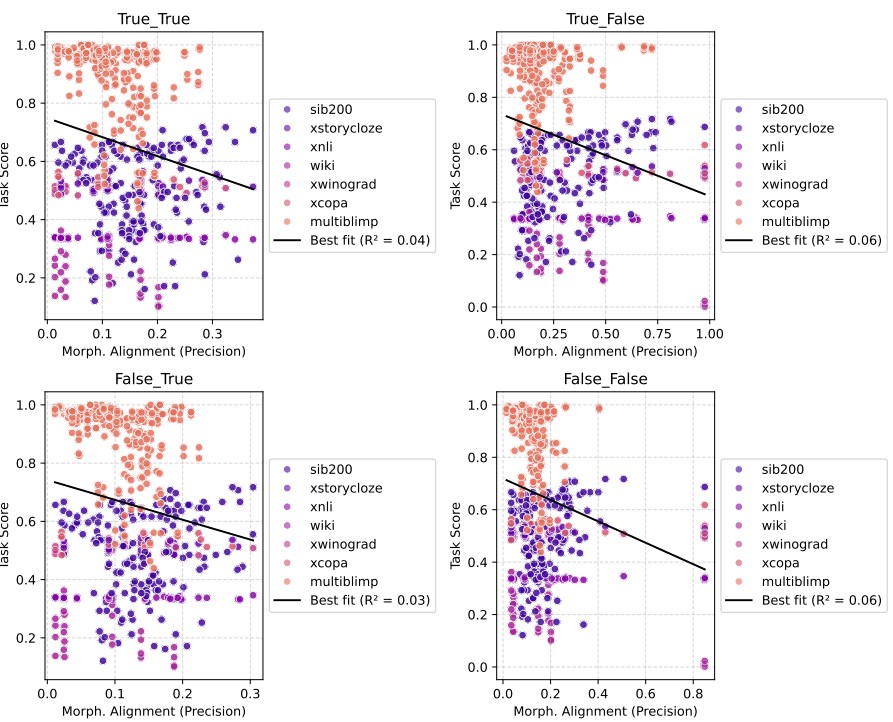

*Figure 3.* Correlation between morphological alignment measured with precision and task score. Model task is indicated by color.

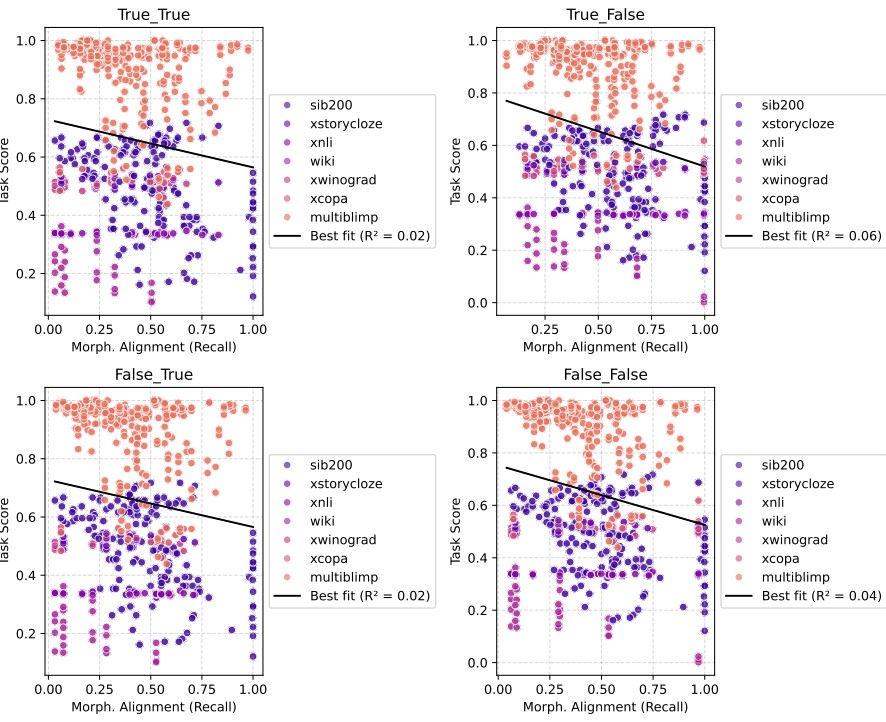

*Figure 4.* Correlation between morphological alignment measured with recall and task score. Model task is indicated by color.

