# OpenReview forum: "Evaluating Morphological Alignment of Tokenizers in 70 Languages"
_ICML.cc/2025/Workshop/TokShop — TokShop_

### Official Review · Reviewer_hjAR · 2025-06-08
**Short paper with great match for the workshop. Useful resource and results, but could benefit from further experiments**

**Rating:** 6
**Confidence:** 4

**Review:**

**Summary:**

This paper measures the morphological alignment of tokenizers in a few popular LLMs across 71 languages, using a dataset compiled and filtered from Universal Dependencies treebanks across 86 languages. The authors propose an evaluation methodology for morphological alignment, expanding on prior work such as MorphScore. Building on their evaluation of morphological alignment, the authors also study the correlation of this alignment with downstream performance on NLP benchmarks, finding that better alignment does not correlate with better performance (if anything, there is a small negative effect).

**Strengths:**

- The role of morphology in LLM performance is still not well understood, in particular for languages beyond English, so the study targets a meaningful goal in addressing this gap
- The findings on the correlation between morphological alignment and downstream performance, albeit a negative result, can help inform the development of better intrinsic evaluation methods and better methods for text representation / tokenization
- The authors compiled a dataset across 86 languages which will be released on huggingface and should be a useful resource for future research in this direction (that being said, it is built on existing resources  — UD specifically — rather than collecting new source data, so the main data contribution lies in the processing)
- The design decisions of the proposed evaluation method are well motivated
- The selection of languages, tasks, and models is generally sensible, and the authors acknowledge limitations arising from this particular selection

**Weaknesses:**

- The finding that morphological alignment does not explain model performance implies that, at least standalone, the alignment score has very limited utility as an intrinsic evaluation
- The paper only presents aggregated results, despite the broad language coverage. It would have been nice to see a breakdown across linguistic features, language family, etc.
- The paper explicitly suggests further experiments e.g. "Combining morphological alignment with other intrinsic tokenizer evaluation metrics, like compression or Rényi efficiency", but leaves them for future work. These experiments could have substantially strengthened the paper, which is anyway well below the page limit, so it would have been nice to directly conduct them (they also don't sound particularly expensive or time-consuming to carry out on top of the work that has been done).
- While such experiments are understandably infeasible for large models, some analyses on models trained with the same architecture and data, i.e., only varying the tokenizer, could help eliminate confounders. It is not clear why the authors exclusively relied on pre-reported performance scores
- Even though the authors include benchmark task as a random intercept to account for mean task difficulty, pooling across tasks with very different formats seems a bit problematic to me since some tasks may benefit more from morphological alignment than others

**Comments:**

Overall, the workshop is a good fit for the paper, and the resources the paper promises to release should be useful to the community. I think it is appropriate as a short paper, but compared to 9-page submissions the paper might lack some experimental depth.

---

### Official Review · Reviewer_xcgK · 2025-06-10
**Super motivation, but the implementation seems suboptimal**

**Rating:** 7
**Confidence:** 3

**Review:**

The paper presents a multilingual evaluation framework for tokenizers
that evaluates whether a given tokenization is "morphologically aligned".
Correlation of this MorphScore and performance on downstream tasks
is computed and the effect of several versions of MorphScore on the correlation is studied.

## Strengths

* Such evaluation framework could help understanding different (multilingual or monolingual) tokenizers.
* The source codes are promised to be published at GitHub and the datasets at HF.
* I appreciate the effort to include many languages (though the chosen uniform approach is not suitable for many languages as described below in Weaknesses).

## Weaknesses

* The Limitations section does not discuss several important limitations.
The proposed "morpheme detection" based on comparing a word form and lemma
is a very crude heuristic. It completely ignores derivations,
e.g. lemma of "actively" is "actively", so the tokenization "active+ly"
(or even "act+ive+ly") is considered wrong.
Similarly with compounds (e.g. in "typewriters" only the final "s" is considered a morpheme).
Thus the precision score may be underestimated in practice
(especially in languages with frequent compounding, such as German).
Also, the UD treebanks are often very small (because they require manual syntactic annotation)
and/or noisy (because the morphological annotation is not required to be manual).
(Using UniMorph as suggested in the Limitations section would not solve all the issues, but it could help.)

* The paper discusses Semitic languages and includes Hebrew,
but it does not discuss the problem of segmentation homographs,
where the same form (e.g. !Pשא) can have multiple segmentations
(e.g. Pַא+¬!שׁ=for+even and !Pַאµשׁ=he aspired).
The paper mentions sentence context of UD as one of the advantages,
but the "frequency scaling" is described as
"Higher-frequency items would be weighted more heavily",
so it seems that even if the homographs were correctly disambiguated in UD,
this information would be lost by merging all words with the same form to a single item.

* UD distinguishes (surface) tokens and (syntactic) words,
which is very relevant (though sometimes tricky) for segmentation.
This aspect is not discussed, so it seems multi-word tokens are completely ignored.

* The sentential context is mentioned several times,
but it is not clear whether it is actually used/needed.
Word frequencies can be easily obtained from large raw corpora (and added to lexicons).
No superword tokenizers are used.
No Scriptio continua languages are used
("Mandarin Chinese, Thai, and Khmer" are mentioned in the introduction, but not in Appendix  A.)

* Section 2 proposes "macro average boundary precision and recall"
and "micro and macro average subword precision, recall, and F1".
However, I could not find these metrics further in the text.
Obviously only one of them is reported, but it is not clear which one.

* While the cross-linguistic aspect is mentioned in the introduction
and multilingual tokenizers are studied, the final MorphScore is computed
for each language independently on other languages. It is known that
the downstream performance on many tasks (often zero-shot for most languages)
is influenced by the alignability of a given language and the source language (usually English).

## Comments

line 020: "datasets for 86 languages"
This number is mentioned only in the abstract.

line 024: "It has also been used to compare compression
crosslinguistically, by calculating CTC over parallel text"
Yes, but it cannot be used for comparing corpora of different size
(and parallel corpora are not always available).
A possible solution is to use CPT (characters per token) instead.
It can be computed simply as CPT=CCC/CTC
(CCC=corpus character count, i.e. the number of unicode characters in the corpus).

line 101: "For each language, we chose the
largest available treebank"
This is not true. According to https://universaldependencies.org,
e.g. UD_Dutch-LassySmall has 297k tokens, while UD_Dutch-Alpino only 208k tokens.
Moreover, the star-rating of LassySmall (0.499) is higher than Alpino (0.005).
Most importantly, the version of UD used should be specified.

line 102: "and used all available splits (train,
dev, and test)."
What was the motivation for using (leaking) the test set?
When someone tunes a tokenizer on this MorphScore,
there will be no way to assess its overfitting.

line 113: "the gold segmentation will have at least two morphemes"
If form=lemma, the described algorithm based on the longest shared sequence
obviously outputs only a single morpheme.

line 237: "Figure 3. Truex3 condition (check)"
The caption is insufficient.

line 288: "our version of morphological alignment score
explains relatively little variance in model performance,
even after taking into account model size and training data
proportion."
What about including also the "compression" among the fixed effects?
As mentioned in the introduction it is one of the most popular intrinsic metrics,
i.e. it is known it has some effect although not "robust" (Schmidt et al., 2024).
Maybe combining compression and morphological alignment could be more robust.

---

### Decision · Program_Chairs · 2025-06-10

Accept